

# Managing migraine with over-the-counter provision of triptans: the perspectives and readiness of Western Australian community pharmacists

Shaid Booth, Richard Parsons, Bruce Sunderland and Tin Fei Sim

School of Pharmacy and Biomedical Sciences, Curtin University, Perth, WA, Australia

## ABSTRACT

**Background:** Down-scheduling one or more triptans to Schedule 3 (Pharmacist Only Medicine) from Schedule 4 (Prescription Only Medicine) has been debated in Australia for a decade. This study aimed to evaluate the perspectives and readiness of Western Australian (WA) community pharmacists to manage migraine including over-the-counter (OTC) provision of triptans.

**Methods:** Data were collected using a self-administered paper-based questionnaire, posted to a random sample of 178 metropolitan and 97 regional pharmacies in WA. Respondent pharmacists were surveyed regarding: knowledge of optimal migraine treatment as per current guidelines, resources required to appropriately recommend triptans and attitudes and perspective toward down-scheduling. Data were analysed using descriptive statistics and multivariate regression analysis. Pharmacist/pharmacy characteristics influencing readiness were evaluated by assigning respondents a score based on responses to Likert scale questions. These questions were assigned to five domains based on an implementation model and these scores were used in a general linear model to identify demographic characteristics associated with readiness across each domain.

**Results:** A total of 114 of the 275 pharmacies returned useable questionnaires (response rate: 41.5%). The two most commonly recommended first line OTC agents were a combined paracetamol/non-steroidal anti-inflammatory drugs and aspirin (44/104; 42.3% and 22/104; 21.2%, respectively) which provided context to the respondents' knowledge of optimal migraine treatment. Responses to questions in relation to triptans and the warning signs requiring referral were in line with current guidelines, demonstrating respondents' knowledge in these areas. Nevertheless, most respondents demonstrated uncertainty in relation to the pathogenesis of migraine. If triptans were available OTC, 66/107 (61.7%) would recommend them first-line. The majority (107/113; 94.7%) agreed that down-scheduling would improve timely access to effective migraine medication and 105/113 (92.9%) agreed that if triptans were down-scheduled, pharmacists may be better able to assist people in the treatment of migraine. Most respondents agreed that additional training and resources, including a guideline for OTC supply of triptans and the management of first-time and repeat migraine would be necessary if triptans were down-scheduled. No single demographic characteristic influenced readiness across all five domains.

**Discussion:** Pharmacists were knowledgeable regarding triptans and recognised symptoms requiring referral; migraine knowledge could be improved. Pharmacists

Corresponding author
Tin Fei Sim, T.Sim@curtin.edu.au

supported down-scheduling of one or more triptans in Australia, however they highlighted a need for further training and resources to support migraine diagnosis and provision of OTC triptans. Professional pharmacy bodies should consider these findings when recommending drugs suitable for down-scheduling for pharmacist recommendation.

# INTRODUCTION

Migraine is a common and disabling disorder which affects 4.9 million people in Australia, 71% of whom are women, with an estimated direct and indirect costs of approximately AUD 35.7 billion annually (Migraine in Australia Whitepaper, *Deloitte Access Economics, 2018*). Current Australian treatment guidelines recommend simple analgesics with or without antiemetics as first line treatment for an initial migraine attack (*eTG Complete, 2018*). If simple analgesics are ineffective the subsequent steps are low dose orally-administered triptans, high dose orally-administered triptans, and subcutaneously-administered triptans (*eTG Complete, 2018*). Triptans are currently only available on prescription in Australia (*Therapeutic Goods Administration, 2018*). Currently available over-the-counter (OTC) migraine treatments include simple analgesics such as paracetamol and non-steroidal anti-inflammatory drugs (NSAIDs), combination products containing paracetamol and ibuprofen, as well as medicines for the management of migraine-related nausea and vomiting including prochlorperazine and combination products containing paracetamol and metoclopramide (*Therapeutic Goods Administration, 2018*).

Triptans are 5-hydroxytryptamine$_1$ (5-HT$_1$) receptor agonists, displaying highest affinity at the 5HT$_{1B/1D}$ receptor subtypes (*Connor et al., 1997*; *Napier et al., 1999*; *Tfelt-Hansen, De Vries & Saxena, 2000*). Three main mechanisms have been proposed to explain the pharmacological actions of triptans on migraine; constriction of cranial vessels, inhibition of vasoactive neuropeptide release and inhibition of nociceptive neurotransmission within the trigeminocervical complex in the brain stem and upper spinal cord (*Tepper, Rapoport & Sheftell, 2002*; *Tfelt-Hansen, De Vries & Saxena, 2000*). Extensive research has shown that triptans are effective and safe antimigraine drugs; large meta-analyses have found that at marketed doses, all oral triptans offered favourable responses compared to placebo for both short-term and sustained pain-free responses (*Thorlund et al., 2014*; *Derry, Derry & Moore, 2014*; *Bird, Derry & Moore, 2014*), and were well tolerated (*Ferrari et al., 2002*; *Derry, Derry & Moore, 2014*; *Bird, Derry & Moore, 2014*). Rizatriptan 10 mg, eletriptan 80 mg and almotriptan 12.5 mg have been found most likely to provide consistent success (*Ferrari et al., 2002*), while eletriptan has been found most likely to provide sustained pain-free responses (*Thorlund et al., 2014*).

The United Kingdom (UK) was the first country to down-schedule a triptan in 2006, allowing pharmacists to supply packs of two tablets of sumatriptan 50 mg without a

prescription. Sweden, Germany and New Zealand (NZ) followed over the next 2 years. In Australia, the National Drugs and Poisons Schedule Committee (NDPSC; now the Advisory Committee on Medicines Scheduling) first considered a proposal to include sumatriptan 50 mg in packs of two tablets in Schedule 3 (Pharmacist Only Medicine) in June 2005. Between 2005 and 2007, the NDPSC addressed concerns such as the diagnosis of migraine by pharmacists, the ability of triptans to mask symptoms of more serious conditions and interactions with serotonergic medications. However, the committee ultimately rejected the proposal to down-schedule sumatriptan on the basis that there was no perceived public health need for the change, due to the existence of emergency supply provisions. According to the Western Australia's *Medicines and Poisons Regulations 2016*, emergency supply of medicines up to a maximum of 3 days' worth of treatment may be provided by pharmacists without a prescription, provided the situation satisfies a genuine therapeutic need as assessed by the pharmacist based on their professional judgement (*Government of Western Australia, 2016*). The decision is the only rejected down-scheduling proposal in Australia involving a medicine recently reclassified from prescription only to OTC status in multiple markets (*Association of the European Self-Medication Industry, 2017*; *Gauld et al., 2012*). Sumatriptan has not been formally discussed by the committee since February 2007.

Studies have shown that down-scheduling of triptans may lead to an improvement in treatment outcomes and a reduced financial burden for migraines sufferers, employers and the government. Triptans are most efficacious when taken early in the attack (*Cady et al., 2004*, *2000*; *Goadsby et al., 2008*; *Klapper et al., 2004*; *Mathew, Kailasam & Meadors, 2004*; *Scholpp et al., 2004*), however patients often delay treatment, primarily to avoid running out of their prescription triptan (*Landy et al., 2013*). Therefore, improving the accessibility of triptans may result in improved treatment outcomes (*Tfelt-Hansen & Steiner, 2007*). People with migraine have been found to spend more on their healthcare, primarily due to a greater frequency of physician and emergency department visits (*Edmeads & Mackell, 2002*). Removing the requirement for patients to visit a physician to access triptans may therefore reduce the financial burden of migraine for sufferers. Furthermore, a substantial body of research has highlighted the burden of migraine on employers in the form of work loss and reduced productivity (*Burton et al., 2002*; *Ferrari, 1998*; *Hu et al., 1999*; *Von Korff et al., 1998*; *Zhang, McLeod & Koehoorn, 2016*) and the ability of triptans to reduce migraine-related work loss (*Burton et al., 2009*; *Dasbach et al., 2000*). A European study of the economic impact of down-scheduling a triptan estimated total government savings over six countries would reach €75 million annually, accounting for approximately 13% of the overall direct economic burden of migraine (*Millier, Cohen & Toumi, 2013*).

Safety was a major concern associated with down-scheduling triptans both overseas and in Australia (*National Drugs & Poisons Scheduling Committee, 2006*; *Tfelt-Hansen & Steiner, 2007*; *The Lancet Neurology, 2005*). Triptans have been shown to be safe prescription medications, however there is a lack of information regarding OTC use; a search of the literature elicited no articles indicating any adverse outcomes from OTC use of triptans. Nevertheless, research conducted in Northern Ireland which surveyed

community pharmacists in the region, highlighted safety as a primary concern of pharmacists when making clinical decisions regarding OTC provision of medicines, including sumatriptan (*Hanna & Hughes, 2012*).

Australia has historically followed an international trend to down-schedule medicines to OTC availability (*Gauld et al., 2012*) and thus, it is likely that one or more triptans will be reconsidered for down-scheduling in the future. Down-scheduling triptans would represent a broadening of the role of pharmacists in the treatment of migraine and it is currently unknown if pharmacists are ready to perform this additional role and their perspectives towards the provision of OTC triptans. Therefore, to answer the research question of whether Australian pharmacists are ready for down-scheduling of triptans, the overall aim of this study was to evaluate the perspectives and readiness of Western Australian (WA) community pharmacists to manage migraine including OTC provision of triptans. This included assessing the knowledge of pharmacists of optimal migraine treatment based upon current migraine treatment guidelines, identifying the tools/resources pharmacists would desire to confidently and appropriately manage migraine with OTC triptans and identifying pharmacy and pharmacist characteristics that influence readiness to provide OTC triptans.

Assessing the readiness of pharmacists for implementing practice change is difficult owing to the lack of a validated tool. Previous studies evaluating how ready pharmacists are to implement a new service have typically evaluated factors such as confidence and knowledge (*Thornton et al., 2017*; *Ung et al., 2017*). Although there is no validated tool to assess the readiness of pharmacists to implement a change in practice, there have been models developed to describe factors that hinder or facilitate the implementation of a new pharmacy service. Such a model was developed by *Garcia-Cardenas et al. (2018)*, who described five domains under which these factors can be categorised, namely professional service, pharmacy staff, pharmacy, local environment and system. In the present study, these domains were used to group survey questions to enable readiness to be evaluated.

## MATERIALS AND METHODS

This study used a self-administered postal questionnaire which was developed based on the study objectives, existing literature and guidelines regarding migraine, treatments (*eTG Complete, 2018*), triptans (*Australian Medicines Handbook, 2018*) down-scheduling (*Tfelt-Hansen & Steiner, 2007*) and effective questionnaire design (*Boynton, 2004*). The drafted questionnaire was face and content validated by six academic colleagues with community pharmacy experience and feedback informed the development of the final questionnaire. This study was approved by the Human Research Ethics Committee of Curtin University (HRE2018-0072). Completion and return of the questionnaire was taken as consent to participate in the study.

The final version of the questionnaire consisted of four main sections: Section A: Demographics, Section B: Migraine, Section C: Treatment Options and Section D: Attitudes Towards Down-Scheduling to Schedule 3 (Table 1). Section A consisted of questions that required participants to select one option, Sections B and D included statements to which participants were asked to indicate their opinions using a 5-point

**Table 1  Questionnaire design and justifications for variables.**

| Section | Variables | Related study objective (s) |
|---|---|---|
| Section A: Demographics | Participants' gender, age, number of years' experience, size and type of pharmacy | Identified characteristics that influenced readiness of pharmacists |
| Section B: Understanding of migraine | Signs/symptoms of migraine, causes/triggers of migraine | Assessed the knowledge of optimal migraine treatment as per current migraine treatment guidelines |
| Section C: Treatment options | Current first/second line treatment, treatment most commonly recommended/requested | |
| Section D: Attitudes toward down-scheduling | Attitudes and perspective toward down-scheduling, tools/resources/training requirements, needs and demand of consumers | Identified the tools/resources needed to appropriately manage migraine with OTC triptans |

Likert scale, in which '1' indicated 'strongly agree' and '5' indicated 'strongly disagree'. Section C consisted of both questions that asked participants to select one or more boxes and statements that required responses using a 5-point Likert scale. Demographic information of respondents, included whether or not they were an accredited pharmacist. Accredited pharmacists are pharmacists accredited by either the Australian Association of Consultant Pharmacy or the Society of Hospital Pharmacists of Australia to undertake government-funded medication reviews. The questionnaire is provided as a Supplemental File to this manuscript.

## Sampling and data collection

A stratified proportional sample of 275 WA community pharmacies was obtained from a sampling frame of 459 metropolitan (Greater Capital City Statistical Area) and 162 regional (rural or remote) community pharmacies, based on postal codes, available from the Pharmacy Registration Board of Western Australia (PRBWA) premises register in February 2018. Hospital pharmacies were excluded from the sample population as Australian hospital pharmacists do not routinely provide primary or self-care services to general members of the public, unless they are inpatients of the hospital, which is beyond the scope of the present study. A random selection of pharmacies was obtained using Microsoft Excel's random number generator. A total of 178 metropolitan and 97 regional pharmacies were selected to receive the survey. The total number of 275 was based on an expected response rate of 40% to achieve within a 95% confidence interval, a 10% precision of any characteristic analysed. Strategies to maximise the response rate and reduce non-response bias were undertaken, which included reminders and follow up processes, simplifying the process to return completed questionnaires, as well as careful planning and validation of the questionnaire to produce a questionnaire tool that was succinct and unambiguous.

Survey packages which included the questionnaire, a participant information sheet and a reply-paid envelope, were posted on 9 March 2018 to be returned by 29 March 2018. The questionnaires were addressed to the pharmacy. The questionnaires were coded to allow identification of non-responding pharmacies for follow up purposes. On 6 April 2018, the 229 non-responding pharmacies were identified and posted the same package

and an additional cover letter explaining the significance of this study. Non-responders as of 16 April 2018 were followed up via telephone calls. Upon calling the non-responding pharmacies, requests were received to email a copy of the survey, which was fulfilled; 59 non-responding pharmacies were also emailed the survey. These pharmacies were also given the option to return the survey via email by 23 April 2018. Nevertheless, responses received prior to 11 May 2018 were included in the study analysis, to maximise response rate as previously discussed.

## Data analysis

Data from all sections were entered into an Excel spreadsheet by SB and checked by TFS. Data were then summarised and analysed using simple descriptive statistics (frequencies and percentages) by Excel or the Statistical Package for Social Sciences (SPSS, version 23). A general linear model (GLM) was used to identify any relationships between pharmacy/pharmacist characteristics and responses to questions. To analyse participant's readiness, questions were classified into five groups based on a model proposed by *Garcia-Cardenas et al. (2018)*. The actual allocation of individual questions to the groups was made by an iterative process to achieve a unanimous decision by the authors. Individual questions were able to be allocated to more than one domain if appropriate. The groups corresponded to the five following domains:

- Domain 1: New Professional Service: included statements assessing knowledge of migraine and triptans, opinions on OTC provision of triptans, opinions on tools and resources required to supply triptans OTC and opinions on potential outcomes of down-scheduling triptans to Schedule 3.
- Domain 2: Pharmacy Staff: included statements assessing knowledge of trigger points for referral to a doctor, knowledge of triptans, opinions on OTC provision of triptans, and opinions on migraine diagnosis by a pharmacist.
- Domain 3: Pharmacy: included statements assessing opinions on training and resources required to diagnose migraine and supply triptans OTC.
- Domain 4: Local Environment: included statements assessing knowledge of trigger points for referral and ability of pharmacists to appropriately refer patients to a doctor, opinions on migraine diagnosis by a pharmacist and opinions on potential outcomes of down-scheduling triptans to Schedule 3.
- Domain 5: System: included statements assessing public health need for down-scheduling triptans, suitability of triptans for down-scheduling to Schedule 3 and potential outcomes (including economical outcomes) of down-scheduling triptans to Schedule 3.

Participants were assigned a score based on their responses to Likert scale questions assigned to each domain and organised so that a high score indicated stronger knowledge, confidence in managing migraine or agreement that triptans may be used by pharmacists. Each domain score was then used as a dependent variable in a GLM to identify which, if any, demographic or pharmacy characteristic variables were associated with them.

In a similar manner, sets of questions indicating 'knowledge' of migraine and triptans (14 questions) and 'attitude towards down-scheduling triptans' (14 questions) were identified. For each question, respondents gained one point for correct knowledge or their support and points were accumulated for each of these two factors. The factors were then analysed using a GLM in a manner similar to that used for the domains. For all statistical tests, a $p$-value of ≤0.05 was used to indicate a statistically significant association.

## RESULTS

A total of 114 of the 275 pharmacies returned useable questionnaires between 13 March 2018 and 11 May 2018, resulting in an overall response rate of 41.5%. A total of 81 questionnaires were returned from metropolitan pharmacies ($n = 178$; 45.5%) and 33 questionnaires were returned from regional pharmacies ($n = 97$; 34.0%). A Chi-squared test revealed no difference between the metropolitan and regional response rates ($p = 0.065$). A total of 192 pharmacies were successfully contacted via telephone calls during follow up (13 pharmacies were not able to be contacted by the telephone numbers listed on the PRBWA premises register after two attempts). Demographic data for the respondents and their pharmacies are summarised in Table 2.

Males accounted for 52.6% of respondents and 73.9% were proprietors (sole or partner). Most community pharmacies were located near a doctor's surgery or clinic.

Responses to questions evaluating pharmacists' preferred OTC treatment options are summarised in Table 3.

Pharmacists would commonly recommend metoclopramide when treating migraine OTC (93/110; 84.5%). Opioids were the medication/class of medication most often requested by patients for the treatment of migraine OTC (50/108; 46.3%).

More than half responded that they did not supply triptans as an emergency supply (67/113; 59.3%). Emergency supplies were provided up to twice monthly from 34/113 (30.1%) respondents, three to four times monthly from 8/113 (7.1%) and more than five times monthly from 4/113 (3.5%).

### Knowledge of migraine and triptans

Responses to statements evaluating pharmacists' knowledge about migraine are summarised in Fig. 1.

Most pharmacists (93/112; 83.0%) perceived that *migraine is caused by the vasodilation of cranial vessels* and a large proportion of respondents (73/109; 67.0%) selected 'don't know/unsure' about dysfunction of a brain stem nuclei. The majority of pharmacists do not consider that people with migraine are more likely to experience serious comorbidities. Almost all pharmacists (111/112; 99.1%) would refer children younger than 12 years of age with migraine, patients who have had migraine for more than 72 h and patients who have had a recent head injury and are requesting treatment for migraine, to a doctor.

Pharmacists' knowledge and opinions of triptans are summarised in Fig. 2.

Most respondents strongly agreed or agreed that triptans relieved migraine pain (104/113; 92.0%), however less than half agreed that triptans alone reduced nausea and vomiting associated with migraine (54/111; 48.6%). Most agreed that triptans were most

**Table 2 Demographic data of respondents and pharmacy characteristics ($n$ = 114).**

| Variable | Category | $n$ | (%) |
|---|---|---|---|
| Age (years) | 21–30 | 30 | (26.3) |
| | 31–40 | 46 | (40.4) |
| | 41–50 | 17 | (14.9) |
| | 51–60 | 16 | (14.0) |
| | 61+ | 5 | (4.4) |
| Gender | Male | 60 | (52.6) |
| | Female | 53 | (46.5) |
| | Other/prefer not to say | 1 | (0.9) |
| Years practising as a pharmacist in Australia | <6 | 29 | (25.4) |
| | 6–20 | 60 | (52.6) |
| | >20 | 25 | (21.9) |
| Principal role in the pharmacy | Sole proprietor | 14 | (12.3) |
| | Partner proprietor | 32 | (28.1) |
| | Pharmacist in charge | 29 | (25.4) |
| | Manager | 16 | (14.0) |
| | Employee pharmacist | 21 | (18.4) |
| | Other | 2 | (1.8) |
| Size of pharmacy (turnover) | Small (≤$2 m per annum) | 64 | (56.1) |
| | Large (>$2 m per annum) | 46 | (40.4) |
| | Unanswered | 4 | (3.5) |
| Setting of pharmacy | Isolated | 14 | (12.3) |
| | Shopping strip | 37 | (32.5) |
| | City centre | 4 | (3.5) |
| | Medical centre | 24 | (21.1) |
| | Small shopping centre (15–50 shops) | 26 | (22.8) |
| | Large shopping centre (>50 shops) | 8 | (7.0) |
| | Other | 1 | (0.9) |
| Location of pharmacy | City | 12 | (10.5) |
| | Suburb | 69 | (60.5) |
| | Rural | 30 | (26.3) |
| | Remote | 3 | (2.6) |
| Location of pharmacy in relation to nearest doctor's surgery or clinic | Co-located | 35 | (30.7) |
| | ≤100 m | 32 | (28.1) |
| | 101–500 m | 23 | (20.2) |
| | 501 m–1 km | 17 | (14.9) |
| | >1 km | 7 | (6.1) |
| Accredited pharmacist status | Yes, accredited pharmacist | 33 | (28.9) |
| | Yes, undergoing accreditation | 4 | (3.5) |
| | No, not an accredited pharmacist or undergoing accreditation | 77 | (67.5) |
| Personal history of migraine | Yes | 14 | (12.3) |
| | No | 100 | (87.7) |

**Table 3 Pharmacists' preferred OTC treatment for migraine.**

| Variable | Category | n | (%) |
|---|---|---|---|
| Q13. Current first line recommendation (n = 104) | Paracetamol | 13 | (12.5) |
| | Aspirin | 22 | (21.2) |
| | Other NSAIDs | 9 | (8.7) |
| | Combined paracetamol/NSAID | 44 | (42.3) |
| | Combined paracetamol/metoclopramide | 13 | (12.5) |
| | Refer to a doctor | 1 | (1) |
| | Other | 2 | (1.9) |
| Q14. Current recommendation if first line treatment was contraindicated or did not work (n = 109) | Paracetamol | 24 | (22) |
| | Aspirin | 14 | (12.8) |
| | Other NSAIDs | 16 | (14.7) |
| | Combined paracetamol/NSAID | 27 | (24.8) |
| | Refer to a doctor | 25 | (22.9) |
| | Other | 3 | (2.8) |
| Q15. First line recommendation if triptans were available OTC (n = 107) | Paracetamol | 5 | (4.7) |
| | Aspirin | 9 | (8.4) |
| | Other NSAIDs | 3 | (2.8) |
| | Combined paracetamol/NSAID | 22 | (20.6) |
| | Triptans | 66 | (61.7) |
| | Other | 2 | (1.9) |

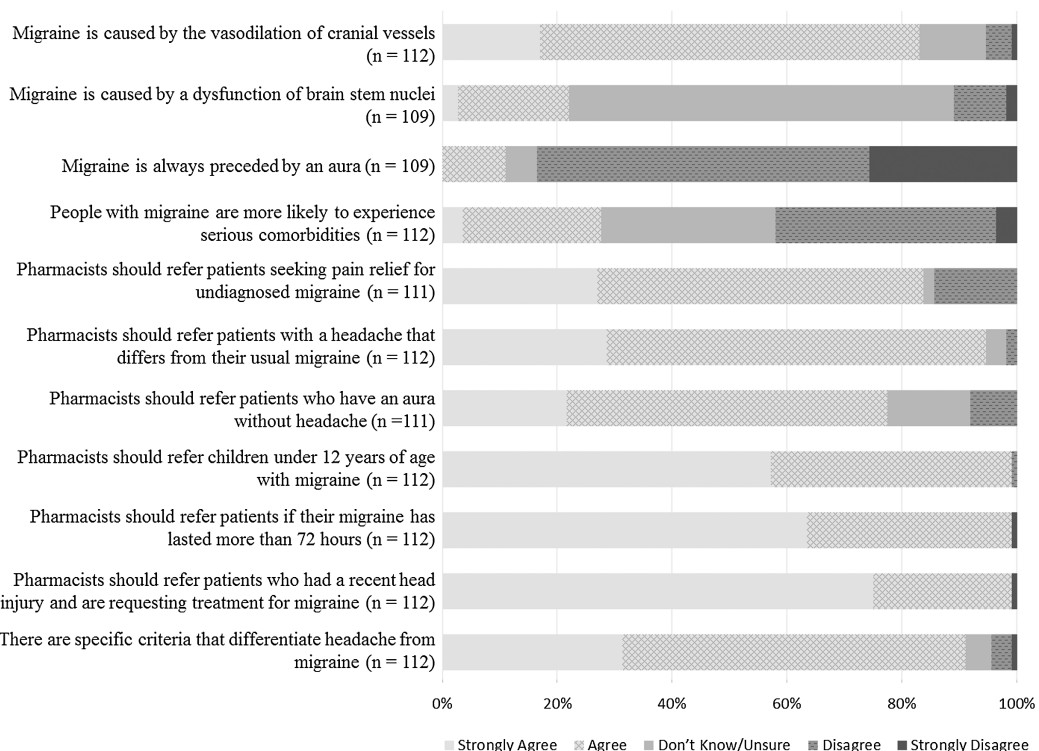

**Figure 1 Respondents' responses to questions regarding their understanding of migraine.**

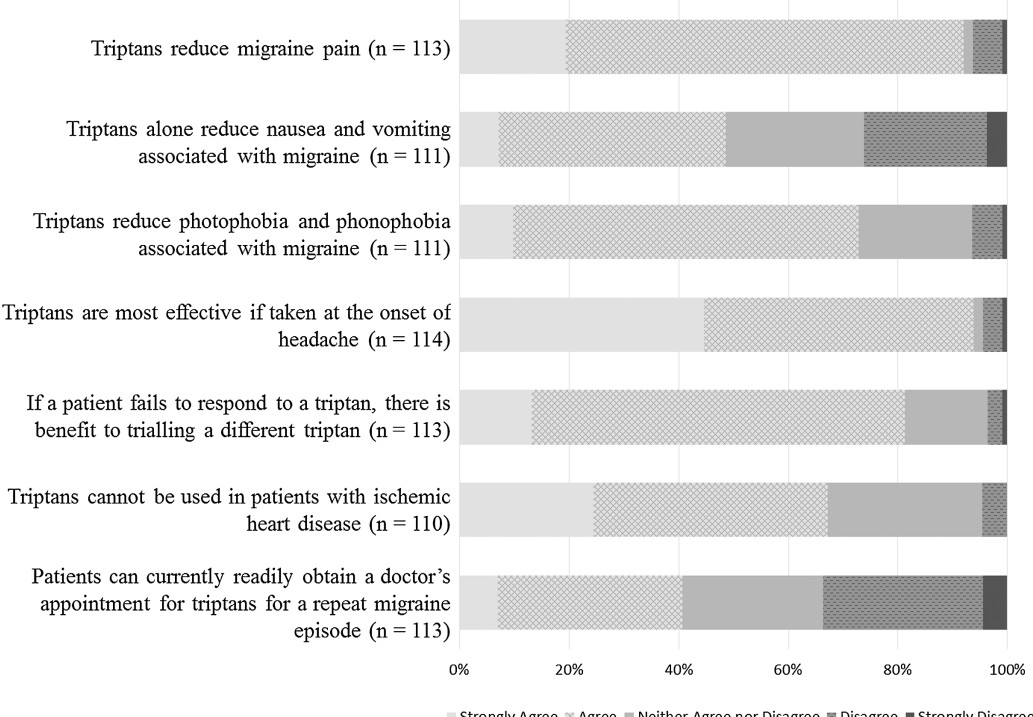

**Figure 2 Respondents knowledge and opinions about triptans.**

effective if taken at the onset of migraine (107/114; 93.9%) and that there was benefit in trialling a different triptan if the patient failed to respond to another (92/113; 81.4%). Responses to the statement *'Patients can currently readily obtain a doctor's appointment for triptans for a repeat migraine episode'* were divided: 40.7% agreement, 25.7% neutral, 33.6% disagreement ($n$ = 113).

The mean score for the 14 questions assessing knowledge of migraine and triptans was 10.9/14 (range: 2–14; SD: 2.1). Although respondents 51 years and above scored less on knowledge questions than respondents from other age groups ($p$ = 0.0020, 0.0311 and 0.0231 for age groups 21–30 years, 31–40 years and 41–50 years, respectively), a low $R$-square value (0.088384) indicated that the demographics of respondents did not largely influence their responses to questions assessing their knowledge of migraine and triptans.

## Resources and training

A total of 111 respondents, 98 (88.3%) and 79 (71.2%) strongly agreed/agreed that pharmacists would require additional training to manage first-time and repeat migraine OTC, respectively. The majority of respondents also strongly agreed/agreed that pharmacists would require additional training to diagnose first time (100/111; 90.1%) and repeat (77/110; 70.0%) migraine. Only 27/111 (24.4%) and 41/111 (36.9%) respondents strongly agreed/agreed that there were sufficient resources to support first time and repeat migraine diagnosis, respectively.

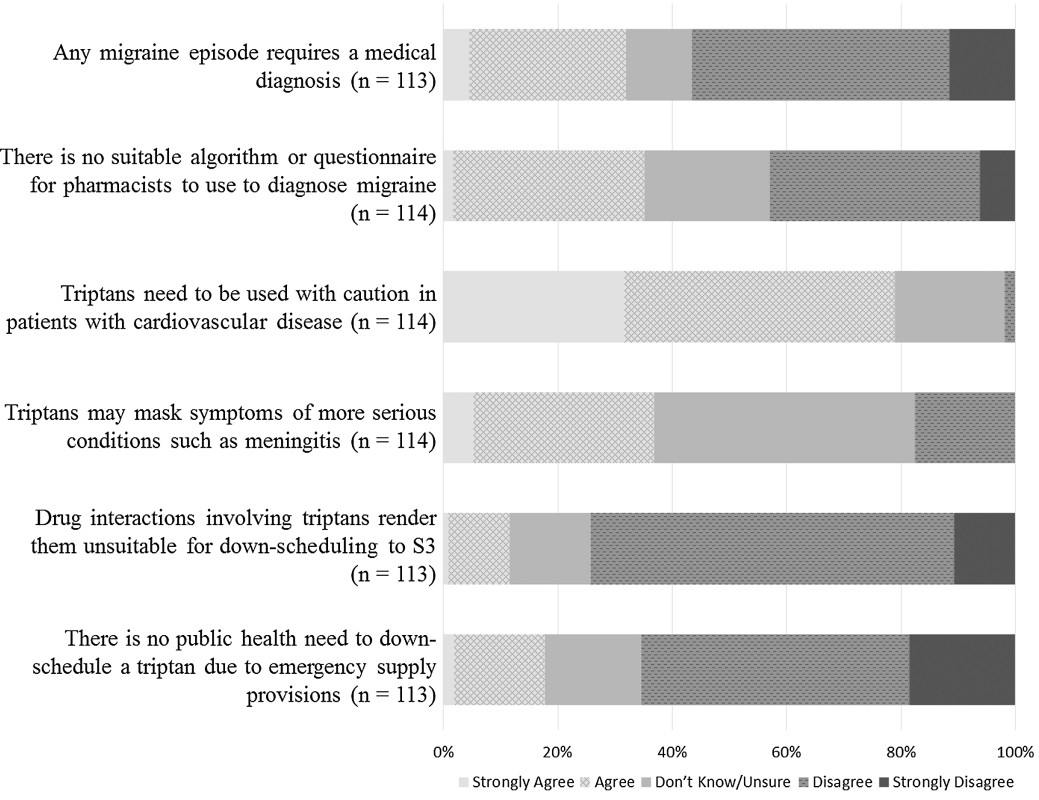

**Figure 3 Responses to reasons put forward by the National Drugs and Poisons Scheduling Committee for not down-scheduling sumatriptan.**

## Attitudes toward down-scheduling

Participants' responses to reasons proffered by the Australian National Drug and Poisons Scheduling Committee (NDPSC) are summarised in Fig. 3.

Pharmacists did not agree with the reasons given by the NDPSC for the rejection of the proposal to down-schedule sumatriptan in 2006–2007 with one exception–respondents agreed that triptans needed *to be used with caution in patients with cardiovascular disease* (90/114, 78.9%). Over one-third of respondents were in agreement with the statement *there is no suitable algorithm or questionnaire for pharmacists to use to diagnose migraine* (40/114; 35.1%). The majority of respondents strongly disagreed or disagreed that *there is no public health need to down-schedule a triptan due to emergency supply provisions* (74/113; 65.5%).

Responses to statements evaluating pharmacists' opinions in relation to OTC provision of triptans are summarised in Fig. 4.

No respondent disagreed with the statement *Triptans should only be available OTC if a pharmacist is involved in the sale.* Most pharmacists strongly agreed/agreed that if triptans were made available OTC, they should only be available in a pack size of two (102/112; 91.1%). A majority of respondents strongly agreed/agreed that triptans are safe to use when provided OTC (68/112; 60.7%).

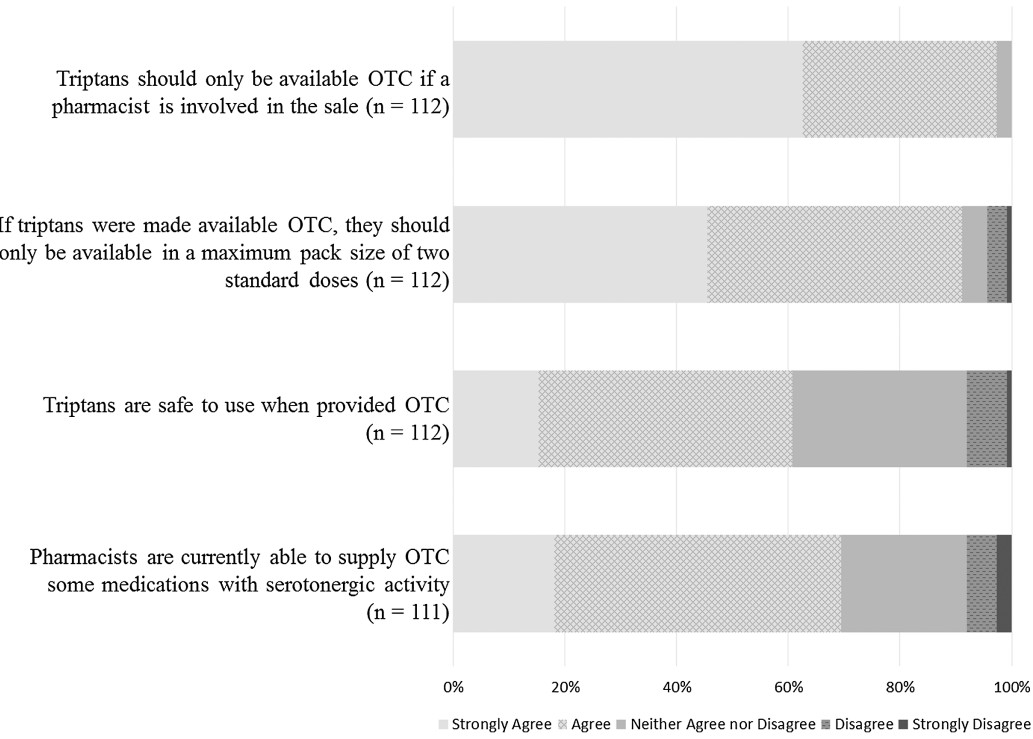

**Figure 4 Opinions on OTC provision of triptans.**

Less than half of the respondents agreed/strongly agreed that most pharmacists can accurately diagnose a first-time migraine (52/112; 46.4%), however a large majority agreed/strongly agreed that most pharmacists can accurately diagnose a repeat migraine (95/111; 85.6%). Most respondents agreed/strongly agreed that doctors can accurately diagnose a first-time (79/111; 71.2%) and a repeat migraine (100/111; 90.1%). Almost all respondents perceived that most pharmacists can accurately identify when to refer patients with migraine for medical review (108/112; 96.4%). Most respondents considered that patients who are migraine sufferers recognise the symptoms of migraine onset (104/111; 93.7%).

Responses to statements evaluating pharmacist's opinions in relation to potential outcomes of down-scheduling of triptans to Schedule 3 are summarised in Fig. 5.

Most respondents strongly agreed/agreed that down-scheduling would improve timely access to effective migraine medication (107/113; 94.7%) and that if a triptan was down-scheduled, pharmacists may be more able to assist people in the treatment of migraine (105/113; 92.9%). Less than half strongly agreed/agreed that down-scheduling would increase the risk of overuse of triptans (47/113; 41.6%), while 36/113 (31.9%) strongly disagreed/disagreed. The majority of respondents either disagreed or strongly disagreed with the statement: 'triptans are too potent for OTC prescribing' (76/112; 67.9%).

The mean score for the 14 questions assessing 'support for down-scheduling was 10.8/14 (range 2–14; SD: 2.8). Scores were influenced by three demographic variables; respondents aged 41–50 years were more supportive of down-scheduling than those aged 21–30 years or 31–40 years ($p$ = 0.0273, 0.0026 respectively), male respondents were more
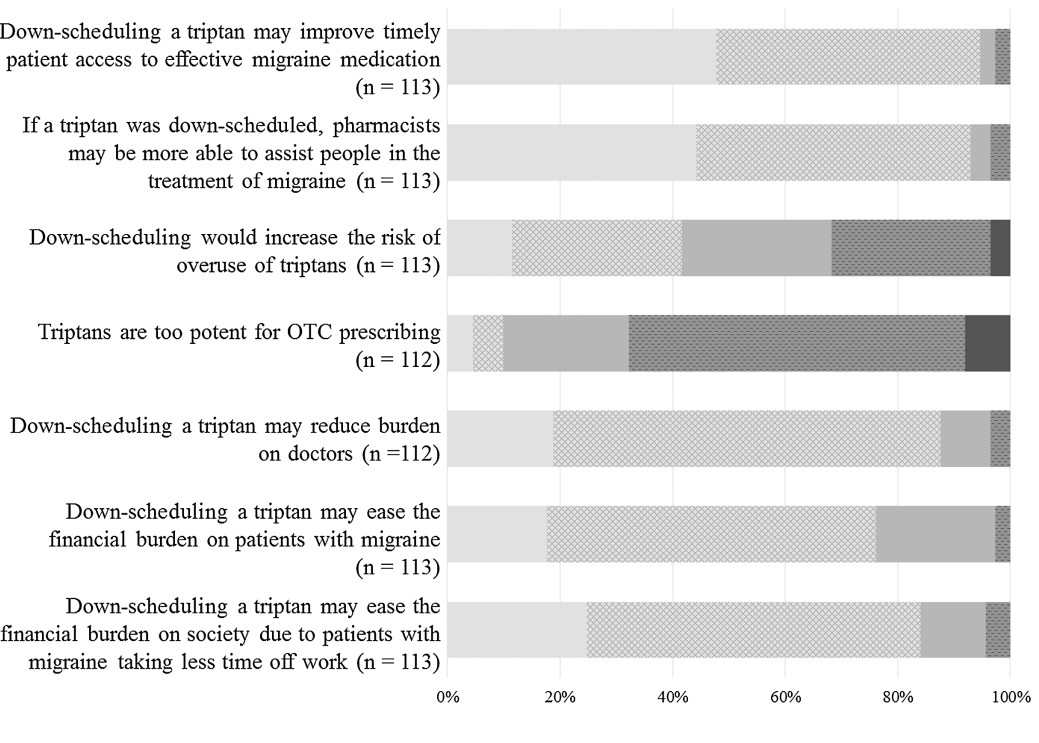

**Figure 5 Respondents' opinions on potential outcomes of down-scheduling of triptans to schedule 3.**

supportive of down-scheduling than female respondents ($p$ = <0.0001) and respondents who suffered from migraine were more supportive of down-scheduling than those who do not suffer from migraine ($p$ = 0.0002).

## Readiness for down-scheduling

For Domain 1: New Professional Service, respondents with fewer than 6 years' experience were significantly 'less ready' compared to respondents with more than 20 years' experience ($p$ = 0.0075). Pharmacists with 6–20 years' experience tended to be 'less ready' compared to pharmacists with more than 20 years' experience, although this association was close to significance ($p$ = 0.0521). Accredited pharmacists, or those in the process of becoming accredited, were significantly more ready compared to pharmacists not accredited/in the process regarding questions in both Domain 1: New Professional Service ($p$ = 0.0184) and Domain 3: Pharmacy ($p$ = 0.0164). Also in Domain 3, pharmacists in the 31–40 years age group were less 'ready' than any other age group ($p$ = 0.0295, 0.0049 and 0.0046 when compared to respondents from the 21–30, 41–50 and 51+ years age groups respectively). There were no demographic variables significantly associated with responses to questions in Domain 2: Pharmacy Staff or Domain 4: Local Environment. Regarding questions under Domain 5: System, employee pharmacists were significantly 'less ready' compared to sole ($p$ = 0.0080) or to partner (0.0070) proprietors and male respondents were significantly more ready compared to female respondents ($p$ = 0.0009).

## DISCUSSION

This study is the first providing information on the perspectives and readiness of WA community pharmacists to manage migraine including OTC provision of triptans. Current OTC management of migraine by pharmacists in this study conforms to Australian treatment guidelines. Pharmacists are generally knowledgeable about triptans and referral points for migraine, however knowledge of migraine could be improved. The results of this study indicate pharmacists would support the down-scheduling of one or more triptans in Australia, however highlights a need for further training and resources to support migraine diagnosis and provision of triptans OTC. The demographic characteristics of respondents influenced aspects of readiness; however, no single demographic characteristic influenced readiness across all five domains.

This study had a stronger male representation than would be expected from the current WA pharmacist workforce statistics. While 52.6% of respondents were male, only 36% of WA pharmacists are males based on 2018 PBA Registrant Data (*Pharmacy Board of Australia, 2018*). This finding is consistent with other survey studies of WA pharmacists; a 2017 study of the views and capabilities of WA community pharmacists regarding the rescheduling of selected antibiotics had 51.1% male respondents, while a 2013 study evaluating the reclassification of ophthalmic chloramphenicol in WA community pharmacies had 44.5% male respondents. The higher male representation may be explained by the overrepresentation of proprietors in survey studies (as the proprietor is often responsible for the mail). In this study, the majority of proprietor respondents were male. Data on the WA community pharmacist workforce was not available for other demographic characteristics, however the age distribution of respondents mirrored those of the national pharmacist workforce.

The results of this study suggest that the current provision of OTC medication for migraine by pharmacists is within recommended guidelines (*eTG Complete, 2018*), with the most commonly selected treatment being combined paracetamol and NSAID. If first line treatment was contraindicated or did not work, approximately 20% of pharmacists would refer to a doctor (compared with just one respondent who would initially refer). This increase in referral rate may reflect adherence to current guidelines as the recommendation is to use a triptan if the first line option is not effective (*eTG Complete, 2018*) and therefore patients need to see a doctor for a prescription to access a triptan.

One interesting finding was that if triptans were available OTC, the majority of pharmacists would recommend them first-line. This finding is outside the guideline recommendations for the initial treatment of migraine; the Therapeutic Guidelines recommends trialling a non-opioid analgesic first and if unsuccessful, to prescribe a triptan for use when the patient next has a migraine (*eTG Complete, 2018*). However, this question did not specify if the migraine was a first-time or repeat migraine and triptans are the recommended first line treatment for repeat migraine where a non-opioid analgesic was previously ineffective.

This study also aimed to identify the training and resources WA community pharmacists would need to confidently and appropriately manage migraine with OTC

triptans. Overall, surveyed pharmacists were knowledgeable about triptans and can correctly identify triggers for referral of migraine patients to a doctor, however their knowledge of the pathophysiology of migraine and common comorbidities was incomplete. The current literature suggests that while vasodilation of cranial vessels does occur in migraine, the cause of migraine pain is due to the activation of trigeminovascular pathways in the brain stem and diencephalic nuclei (*Akerman, Holland & Goadsby, 2011*; *Bernstein & Burstein, 2012*; *Goadsby et al., 2017*). A reasonable explanation for this finding may be that pharmacists have not kept up to date with advancing knowledge regarding migraine pathophysiology, an explanation consistent with a study that found the majority of pharmacists had not completed any continuing education on headaches over a 2-year period (*Wenzel et al., 2005*). Therefore migraine-focused continuing education sessions may improve the ability of pharmacists to confidently and appropriately manage migraine with OTC triptans.

Furthermore, this study identified a lack of resources available to Australian pharmacists to support the diagnosis and management of migraine. In the UK and NZ, pharmacists can diagnose migraine and supply a triptan where appropriate to patients with a 'well-established pattern of symptoms', provided they use a validated tool, that is the Migraine Questionnaire (*Medicines & Healthcare Products Regulatory Agency, 2006*). Furthermore, the Royal Pharmaceutical Society released a 'quick reference guide' when sumatriptan was down-scheduled in the UK, which provided criteria for sumatriptan supply, precautions and contraindications for use, counselling points and further references (*Royal Pharmaceutical Society, 2006*). The manufacturer of Imigran Recovery™ (an OTC-branded sumatriptan in the UK) also launched a National Pharmacy Association-accredited training resource for pharmacy staff in 2012 (*Brown, 2012*). The Pharmaceutical Society of Australia has developed such documents for other medications down-scheduled from Schedule 4 to Schedule 3 in Australia including chloramphenicol eye drops, proton-pump inhibitors and emergency contraceptive pills (*Pharmaceutical Society of Australia, 2018*). If a triptan was to be down-scheduled in Australia, the results of the present study indicate resources such as a Migraine Questionnaire and relevant guidance documents be part of any down-scheduling decision.

Support for down-scheduling triptans was assessed in a number of ways, including opinions on: reasons given by the NDPSC when rejecting previous proposals to down-schedule sumatriptan, the suitability of triptans for OTC use and potential outcomes of down-scheduling a triptan. Most pharmacists did not agree with the reasons given by the NDPSC for the rejection of the proposal to down-schedule sumatriptan in 2006–2007 with one exception—a majority of respondents agreed that *triptans need to be used with caution in patients with cardiovascular disease*. This response is in line with current Australian data (*Australian Medicines Handbook, 2018*).

The results of this study indicate that pharmacists consider triptans suitable for OTC use. Pharmacists' overall support for down-scheduling of triptans as demonstrated in this study, were in contrast to the views of the NDPSC in 2006–2007. However, pharmacists showed some concern regarding triptan overuse—over 40% agreed that down-scheduling would increase the risk of overuse of triptans and the issue of overuse

was a common theme in respondents' additional comments where provided. Studies conducted in America and some European counties (all of which have triptans available on prescription only) have found that triptan overuse occurs in up to 10% of patients and contributes to medication overuse headache (*Braunstein et al., 2015*; *Da Cas et al., 2015*; *Dekker et al., 2011*; *Schwedt et al., 2018*). To the authors' knowledge, no studies have been done to evaluate the effect of down-scheduling a triptan on the rates of triptan overuse.

Three demographic characteristics were associated with greater support for down-scheduling: age, gender and migraineur status. Respondents aged 41–50 years were more supportive of down-scheduling than those aged 21–30 years or 31–40 years. This finding may be explained by younger pharmacists having less confidence than older pharmacists, or older pharmacists having had more time to build rapport with regular patients. Male respondents were more supportive of down-scheduling than female respondents. Respondents who suffered from migraine were more supportive of down-scheduling than those who do not suffer from migraine which could be expected due to the fact that most respondents agreed that down-scheduling would improve timely access to effective migraine medication.

In regard to the ability of pharmacists to collaborate with other health professionals in the treatment of migraine, almost all respondents perceived that most pharmacists can accurately identify when to refer patients with migraine for medical review; no respondents disagreed/strongly disagreed with this statement. This response is supported by the large majority of participants that correctly identified trigger points for referral. While a previous study found the majority (54%) of pharmacists were comfortable with their ability to identify patients with migraine needing physician referral (*Wenzel et al., 2005*) the current study had a much larger majority of respondents in agreement (96.4%).

Some demographic characteristics of pharmacists influenced their responses to questions assessing readiness over three of the five domains, though low $R$-squared values indicated that the demographics of respondents did not largely influence their responses. Pharmacists with more than 20 years' experience and accredited pharmacists or those undergoing accreditation were 'more ready' within Domain 1: New Professional Service. This finding could be expected given pharmacists with more experience and those who have undergone further training are more likely to have experience in implementing a new service. Accredited pharmacists or those undergoing accreditation were also 'more ready' regarding Domain 3: Pharmacy. Pharmacists in the 31–40 years age group were 'less ready' than any other age group regarding questions under Domain 3: Pharmacy. Regarding Domain 5: System, employee pharmacists were 'less ready' compared to sole or partner proprietors. As Domain 5 included questions relating to policy, legislation and economics, this finding could be explained by the additional experience that proprietors have in these areas. Male respondents were 'more ready' compared to female respondents regarding Domain 5: System, however, as 73.9% of the proprietors were male, this finding can be expected given proprietors also indicated higher readiness regarding the questions in this domain.

There were no demographic variables significantly associated with responses to questions in Domain 2: Pharmacy Staff or Domain 4: Local Environment. Responses to the

questions in these domains were not based upon pharmacist or pharmacy variables. Furthermore, there were no demographic characteristics consistently associated with readiness scores across all five domains, which suggests that although some characteristics of pharmacists may influence aspects of readiness, overall readiness to supply OTC triptans was not greatly influenced by demographic characteristics.

This study has several limitations. The response rate of 41.5% was as predicted but does not ensure that non-respondents had similar views. There is no known reason why these would be different, especially when many of the findings were clear. Respondents could have looked up answers to knowledge questions but that is unlikely in this type of survey, especially as respondents are busy. The small sample size of certain demographic groups (e.g. pharmacists aged 61+ years) restricted multivariate regression analysis. Furthermore, the model used to evaluate pharmacy and pharmacist characteristics that influence readiness was published as a theoretical model of factors influencing the implementation of professional pharmacy services and has therefore not been validated as a tool to determine readiness. This approach did not allow for easy assessment of the general readiness of the group as each domain was scored separately. However, the questionnaire was designed to encompass the factors that were reported to influence readiness of pharmacists. The development or validation of a model to assess readiness would be advantageous in further studies aiming to assess readiness of pharmacists.

Although not within the scope of this study, it is notable that sumatriptan is the only triptan considered for down-scheduling in Australia (and the only triptan available without a prescription in the UK) despite literature suggesting it is not the most effective triptan. Meta-analyses of all marketed triptans suggest the triptans most likely to produce consistent success are rizatriptan, eletriptan and almotriptan (*Ferrari et al., 2002*) and that eletriptan is the triptan most likely to produce sustained pain-free responses (*Thorlund et al., 2014*). If one or more triptans are to be considered for down-scheduling in Australia, further consideration is necessary to identify the triptan(s) most appropriate for OTC provision.

It is also important to consider the potential impact of triptan down-scheduling, taking into consideration international experience. A qualitative study by *Paudyal et al. (2013)* published in 2013 explored pharmacists' adoption of newly down-scheduled (or re-classified) medicines in the UK. It was reported that whilst strategies to enable safe supply of reclassified medicines were necessary, the risk assessment tools, including comprehensive questionnaires for the supply of sumatriptan, were regarded as a barrier (*Paudyal et al., 2013*). Another study explored pharmacy students' perspectives on OTC medicines, including triptans and identified that restrictive product licences and manufacturers' restrictions a barrier to self-care (*Hanna, Hall & Duffy, 2016*).

Whilst this study focuses on the management of migraine with OTC provision of triptans, the questionnaire and study protocol may be adapted to assess pharmacists' readiness for down-scheduling of other medicines and in the management of other medical conditions, for example antibiotics for urinary tract infection, combined oral

contraceptives for contraception and 5-phosphodiesterase inhibitors for erectile dysfunction.

## CONCLUSIONS

This study has found strong support from respondents for the down-scheduling of triptans for better management of migraine by community pharmacists. There was evidence in some of the domains that males, pharmacists with more than 20 years' experience or those who were accredited were the most ready for this change, while pharmacists in the 31–40 years age group and employee pharmacists were less ready, however, no demographic characteristics were associated with a higher readiness score across all five domains. The results of this study also indicate that pharmacists currently manage migraine according to guidelines and refer patients appropriately. Despite WA pharmacists' readiness to manage migraine with OTC triptans, implementation is not possible until appropriate amendments are made to legislative, scheduling and manufacturing restrictions. There would be benefits to patients and society for triptans to be down-scheduled to 'Pharmacist Only Medicine' status. Professional pharmacy bodies in Australia should consider these findings when considering down-scheduling of triptans in Australia and the study may form useful background when considering other Schedule 3 medicines.

## ACKNOWLEDGEMENTS

The authors would like to acknowledge all pharmacists who took part in this study.

### Funding

The authors received no funding for this work.

### Competing Interests

The authors declare that they have no competing interests.

### Author Contributions

- Shaid Booth performed the experiments, analysed the data, contributed reagents/materials/analysis tools, prepared figures and/or tables, authored or reviewed drafts of the paper, approved the final draft.
- Richard Parsons performed the experiments, analysed the data, contributed reagents/materials/analysis tools, authored or reviewed drafts of the paper, approved the final draft.
- Bruce Sunderland conceived and designed the experiments, performed the experiments, analysed the data, contributed reagents/materials/analysis tools, authored or reviewed drafts of the paper, approved the final draft.
- Tin Fei Sim conceived and designed the experiments, performed the experiments, analysed the data, contributed reagents/materials/analysis tools, authored or reviewed drafts of the paper, approved the final draft.

## Human Ethics

The following information was supplied relating to ethical approvals (i.e. approving body and any reference numbers):

The Curtin University Human Research Ethics Committee approved the study (HRE2018-0072).

## Data Availability

The raw data are available in the Supplemental Files.

## Supplemental Information

Supplemental information for this article can be found online at http://dx.doi.org/10.7717/peerj.8134#supplemental-information.

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
