# Peer review of "Managing migraine with over-the-counter provision of triptans: the perspectives and readiness of Western Australian community pharmacists"

_PeerJ, doi:10.7717/peerj.8134_

## Round 0.1 · original submission · Major Revisions

Dear authors,
Major revisions are required in your manuscript previous to be accepted. Please, answer point by point each of the questions of each reviewer.
Best
The editor

·

Basic reporting

• Clear and unambiguous, professional English used throughout.

The language is intelligible and professional. There are some parts of the manuscript which could be clearer (aim and parts of the methodology, as mentioned later in this review).


• Literature references, sufficient field background/context provided. .

Line 77 Is there more recent information about migraine prevalence (beyond the 2007 reference)?
Lines 82-83 Perhaps you could be explicit about what is currently available OTC in Australia to manage migraines, rather than just stating that triptans are not. This might help the international readership and provide more context. For example, in addition to simple analgesia and some combined analgesia products, the UK would have buccal prochlorperazine (Buccastem M) licensed for the treatment of nausea and vomiting associated with migraine. Is that available in Australia too? Oral diclofenac was reclassified back to POM in 2015 and buclizine hydrochloride (as an anti-emetic) is limited in usefulness due to being in combination with a codeine-containing product. Metoclopramide is POM is the UK.
Line 90 You could potentially mention Cochrane Systematic Reviews, such as:
o Derry CJ, Derry S, Moore RA. Sumatriptan (all routes of administration) for acute migraine attacks in adults ‐ overview of Cochrane reviews. Cochrane Database of Systematic Reviews 2014, Issue 5. Art. No.: CD009108. DOI: 10.1002/14651858.CD009108.pub2
o Bird S, Derry S, Moore RA. Zolmitriptan for acute migraine attacks in adults. Cochrane Database of Systematic Reviews 2014, Issue 5. Art. No.: CD008616. DOI: 10.1002/14651858.CD008616.pub2
Starting Line 112 While sumatriptan has been available OTC in other countries/regions for several years, is there any evidence that this deregulation has been successful/beneficial? Perhaps such information should/could affect the decision by ACMS to reclassify? When I asked UK pharmacists about it around 10 years ago as part of my doctoral research, they didn’t seem to have favourable opinions on it.
Line 384 “this study is the first providing information on the perspectives and readiness of community pharmacists to manage migraine including OTC provision of triptans”. There is other work that has gleaned information on UK pharmacists’ views on OTC deregulations including sumatriptan [Hanna LA, Hughes CM. Pharmacists’ attitudes towards an evidence-based approach for over-the-counter medication. International journal of clinical pharmacy. 2012 Feb 1;34(1):63-71].
Line 437 - the Royal Pharmaceutical Society of Great Britain (RPSGB) is now known as the Royal Pharmaceutical Society (RPS), not RPSGB. A manufacturer of OTC sumatriptan (Imigran Recovery) also provided support material https://www.chemistanddruggist.co.uk/feature/imigran-recovery


• Professional article structure, figures, tables. Raw data shared.

The suggested format of the manuscript has been followed. It is difficult to check word count in the pdf, but the abstract appears to be around 370 words.
Figures seem clear, the only minor comment is that they could include the question number (not necessarily the question part) like has been done for Table 3.
Raw data/data collection tool: The final version of the instrument (questionnaire) has been uploaded although it’s not clear whether the few sentences at the start is the ‘cover sheet’ in its entirely, or whether more information about the study was provided in the survey package and/or via the telephone call. The raw questionnaire data has been provided within a spreadsheet. This appears to align with the questions in the questionnaire (columns of the spreadsheet) and with the response rate of 114 (number of rows in the spreadsheet). It would be helpful to provide a key too which shows the questionnaire options mapped to numerical coding in the spreadsheet (e.g.
Male = 1, Female = 2, Other =3;
Strongly agree = 5 to Strongly Disagree = 1 etc.
The figures and tables do not appear to have been inappropriately manipulated - it’s difficult to say definitively as there are no frequency totals in the raw data spreadsheet or explanations about what the numerical codes mean.
Figure 2: “Respondents knowledge…” should be “Respondents’ knowledge…”
There is some repetition of results (text and tables); potentially the text could be more succinct or differ from the information presented in the table.

• Self-contained with relevant results to hypotheses.

Abstract Method/results: Some of the results could align a bit more closely to the information provided in the Method. The Method states you were ascertaining: “their knowledge of optimal migraine treatment as per current guidelines, the resources required to appropriately recommend triptans…” Therefore, maybe rephrase the results about the first line agents to be in the context of respondents’ knowledge of optimal treatment and consider including something about their views on what resources would be required to enable an appropriate recommendation of triptans. Abstract Discussion/results: it is difficult to see where the information in the discussion is derived from (assuming the abstract is a stand-alone piece), for example, that they were knowledgeable regarding triptans, recognised symptoms requiring referral, but that their migraine knowledge could be improved. None of these points are really apparent from the results presented.

Experimental design

• Original primary research within Aims and Scope of the journal.

Yes

• Research question well defined, relevant & meaningful. It is stated how research fills an identified knowledge gap.

A research question could be included in addition to the study aim/objectives. The aim could potentially be clearer. You state: “evaluate the…readiness…to manage migraine including over-the-counter (OTC) provision of triptans”. Were you measuring their readiness to manage migraine (in a general sense, which could also include triptans) or their “readiness to manage migraine with triptans” (given these aren’t OTC options currently). Perhaps an explanation or definition about “readiness” would be useful - if readiness encompasses confidence and knowledge (as mentioned on Line 149 in the context of previous work) do you need to have knowledge as a secondary objective? This applies to abstract and main manuscript.

• Rigorous investigation performed to a high technical & ethical standard.

See below – the manuscript could include more information about what the respondents knew about the study prior to participation (in terms of consent/voluntary participation/how their data would be used/stored/whether it was anonymous or could be traced back to a specific pharmacy). Was all of this provided in the survey pack or discussed in the telephone calls? Information about ethical approval is provided within the manuscript on Lines 164-165 i.e. the study was approved by the Human Research Ethics Committee of Curtin University (HRE2018-0072).

• Methods described with sufficient detail & information to replicate.

Line 173 After describing the development of the questionnaire, you could mention that it is provided in the appendix/available on request from the authors.
Line 181 You could also explain why hospital pharmacies were excluded from your study (is it because they don’t stock OTC medicines/wouldn’t be providing self-care advice/medicines?). You could outline various ways you tried to maximize the response rate (and hence reduce non-response bias) from the outset e.g. reminders, incentives, developing a relatively short questionnaire etc.
Line 186 Please expand on what information was provided on the participant information sheet – did it make it clear that participation in the study was voluntary, purpose of the study, estimated time for completion, how their data would be stored/used? Was consent implied or did you have something relating to this on the cover sheet so that you actually obtained written consent?
Lines 188-189 Can you explain more about the coding of the questionnaires for the purposes of targeting non-responders i.e. does this mean that responses were only partially anonymous as the questionnaire could theoretically be linked to a specific pharmacy? And if it could be linked to a specific pharmacy, were all the demographic questions necessary?
Lines 194-195 You state: “Responses received prior to 11 May 2018 were included in the study analysis.” Perhaps you could explain a bit more about this date (i.e. there was an extended deadline to that initially stated on the questionnaire in order to maximize the response rate) since the questionnaire uploaded as supplementary material states “Please kindly return the completed questionnaire in the reply-paid envelope enclosed by the 23rd of April 2018.”

Validity of the findings

• Impact and novelty not assessed. Negative/inconclusive results accepted. Meaningful replication encouraged where rationale & benefit to literature is clearly stated.

You have presented a mixture of positive and negative findings/opinions and openly discussed these. This work is particularly relevant for Australia given that OTC sumatriptan is already deregulated in other parts of the world (including the UK and other parts of Europe and New Zealand for around 10 years). Perhaps you could mention that while this study focuses specifically on migraine and migraine medicines, the questionnaire could adapted for other clinical conditions/potential POM switches (or other suggestions for future research priorities)

• All underlying data have been provided; they are robust, statistically sound, & controlled.

This seems fine. You could clarify whether data entry was done by one person and then accuracy checked by another independently etc. Also, was a script followed when contacting the pharmacies by phone to ensure consistency of approach.

• Conclusions are well stated, linked to original research question & limited to supporting results.

As previously mentioned about the abstract discussion/results: it is difficult to see where the information in the discussion is derived from (assuming the abstract is a stand-alone piece), for example, that they were knowledgeable regarding triptans, recognised symptoms requiring referral, but that their migraine knowledge could be improved. None of these points are apparent from the results presented.
I think you might want to mention that readiness to supply (and adequate support to do so) could still mean that a supply cannot be made due to the manufacturer’s restrictions. For example, the electronic medicines compendium https://www.medicines.org.uk/emc/product/209/smpc outlines the numerous restrictions imposed by the manufacturer in relation to their product Imigran Recovery and it “should only be used where a clear diagnosis of migraine has been made by a doctor or a pharmacist. For pharmacy supply, patients should have an established pattern of migraine (a history of five or more migraine attacks occurring over a period of at least 1 year).” There are numerous cautions and contra-indications too.

• Speculation is welcome, but should be identified as such.

It is clear when the authors are speculating/postulating about their findings.

Additional comments

This was an interesting piece of work and perhaps future research could ascertain opinions of reclassifications (potential and existing) given the drive for self-care and self-medication.

I have undertaken the peer review using the journal's criteria (these are bullet points throughout the review) and provided comments about each.

Reviewer 2 ·

Basic reporting

Overall, this was an interesting paper with information presented logically and coherently throughout. The writing is of a very high standard with only a few typographical errors e.g. lines, 113 and 136. I would suggest that units are used throughout the text when referring to age e.g. line 299, 51 years rather than 51.

The introduction to the paper provides a detailed background to triptan deregulation with appropriate references cited. This section could be improved by providing a summary table of licenced drugs for migraine treatment available OTC or under pharmacist supervision. This would be particularly beneficial for international readers in terms of context. For example, metoclopramide is discussed later in the manuscript, but this is a prescription only medicine in the UK.

No issues with any other aspects of ‘basic reporting’

Experimental design

No comment

Validity of the findings

In general, the findings have been clearly identified and appropriately discussed. There is some restating of results within the discussion (lines 488-503).

Could have further review within the discussion on the potential impact of triptan deregulation. What can be learnt from the experience of other countries? E.g. it has been stated by Paudyal et al. 2013 that the reclassification of sumatriptan in the UK has been not been particularly successful. Another study by Hanna et al. 2016, reported that pharmacy students’ in the UK thought that the product licences were too restrictive and may be a barrier to self-care.
(Vibhu Paudyal, Denise Hansford, Scott Cunningham, Derek Stewart; Over-the-counter prescribing and pharmacists' adoption of new medicines: Diffusion of innovations, Research in Social and Administrative Pharmacy, 9:3, 2013, Pages 251-262)
(Lezley-Anne Hanna, Maurice Hall, Deirdre Duffy; Pharmacy students’ use and views on over-the-counter (OTC) medicines: a questionnaire study, Currents in Pharmacy Teaching and Learning, 8:3, 2016, Pages 289-298)

Supplemental data has been provided, but it is noted that there was an ‘additional comments’ section within the questionnaire. Were the responses obtained analysed? If so, what were the key findings. If not, why was this the case?

Additional comments

Overall, a very interesting paper with insightful and relevant findings.

·

Basic reporting

This is a well written manuscript with clear and unambiguous reporting. Literature used well to set the scene and the structure and figures support the authors' arguments. I particularly liked Table 1 and the mapping of domains to study objectives.

Some minor comments re. presentation (particularly for an international audience):
What is the difference between a metropolitan and a regional community pharmacy?
What is "Accredited pharmacist status"? This needs addressed early in the paper as it became a significant variable in the findings.
What is the process for Emergency supplies and is there any literature to support prevalence of their use in general in Australia?
What is the purpose of metoclopramide i.e. anti-emetic - useful for the non-expert reader?
Line 469 To the author’s knowledge - should be plural.

Experimental design

This study makes an original contribution to knowledge in this area and the study has been robustly designed and executed. Methods are clearly described.

Validity of the findings

Clear rationale for what this study adds to the literature and the limitations. Clearly stated conclusions which in the main are supported by the findings. Minor comments:

Line 385 ... of migraine by pharmacists [in this study!] conforms to Australian treatment

542 ... Professional pharmacy bodies in Australia should consider these findings when considering recommendations for further Schedule 3 medicines to become available in Australia.
In my opinion, this statement is too far reaching, given the findings. Prof bodies should consider these findings when considering down-scheduling of triptans and the study may form useful background when considering other schedule 3 medicines - I would recommend rewording this final statement.

Additional comments

Thank you for the opportunity to review this interesting and well written paper. The research is robust and adds to knowledge in this field. My comments relate mainly to presentation, in particular for an international audience.

---

## Round 0.2 · accepted · Accept

Dear authors,

I am happy to inform you that your paper has been Accepted.

·

Basic reporting

As previously stated, the language is intelligible and professional.
In this revised version:
References have been added that should be useful to the international readership.
Raw data has been shared and this makes more sense now that a key has been included. Structure is fine and figures have clearer labelling.
Results are more succinct and easier to follow, as is the case for the abstract too (aligned to aims)

Experimental design

As previously stated, this work falls within the scope of the journal.
In this revised version:
The research question is clear, relevant and meaningful and it is apparent how this study adds to the existing body of work already conducted in this area.
In terms of rigour and ethical considerations, the aim and methodology have now been amended to a satisfactory level. More information has been provided about what the participants were told prior to agreeing to participate.

Validity of the findings

As previously stated, the authors presented a mixture of positive and negative findings/opinions and openly discussed these. This work is particularly relevant for Australia given that OTC sumatriptan is already deregulated in other parts of the world (including the UK and other parts of Europe and New Zealand for around 10 years).

In this revised edition:
The authors have now included how the questionnaire could be adapted for other clinical conditions/potential POM switches.
The authors have now clarified about data entry and accuracy checking, and provided appropriate underlying data and a coding key.
The abstract is a better stand-alone piece.
The conclusion maps to the research question/study aims and objectives.
In terms of postulating/ speculating, the authors have mentioned issues around the manufacturer’s numerous restrictions for OTC sales (i.e. regardless of the pharmacist's level of readiness)..

Additional comments

Thank you for taking my feedback and suggestions on board; I think the paper has been substantially strengthened now.

Reviewer 2 ·

Basic reporting

Overall, this was an interesting paper with information presented logically and coherently throughout. The writing is of a very high standard and previously identified typographical errors have been rectified.

The introduction to the paper provides a detailed background to triptan deregulation with appropriate references cited. The authors have addressed my previous comment on providing a summary of products currently licensed OTC for migraine management in Australia.

No issues with any other aspects of ‘basic reporting’

Experimental design

Interesting and topical area of research with appropriate methods adopted to meet the identified primary objectives of the study.

Validity of the findings

Findings have been clearly identified and appropriately discussed. My previous comment regarding restating of results in the discussion has been addressed and the manuscript updated accordingly. Authors have also expanded the discussion to consider the potential impact of triptan deregulation and have discussed other relevant studies in the field.
Authors have clarified in their response why results from ‘additional comments’ have not been presented in the manuscript. Their rationale is sound their methods are appropriate.

This is an interesting study which has been very well presented and I anticipate it will be welcomed in the field, with particular relevance for those working in Australia.

Additional comments

I recommend this paper for publication.